# Gram-Level Production of Balanol through Regulatory Pathway and Medium Optimization in Herb Fungus *Tolypocladium ophioglossoides*

**DOI:** 10.3390/jof8050510

**Published:** 2022-05-16

**Authors:** Rui-Qi Li, Xiang Liu, Min Zhang, Wei-Qun Xu, Yong-Quan Li, Xin-Ai Chen

**Affiliations:** 1School of Medicine and the Children’s Hospital, Zhejiang University, Hangzhou 310058, China; 21918020@zju.edu.cn (R.-Q.L.); 22018007@zju.edu.cn (X.L.); zjuzm@outlook.com (M.Z.); vicky__xu@zju.edu.cn (W.-Q.X.); 2Institute of Pharmaceutical Biotechnology, Zhejiang University, Hangzhou 310058, China; lyq@zju.edu.cn

**Keywords:** balanol biosynthesis, protein kinase C inhibitor, Zn_2_Cys_6_, regulator BlnR, medium optimization, fermentation, *Tolypocladium ophioglossoides*

## Abstract

As a potential protein kinase C inhibitor, the fungus metabolite balanol has become more attractive in recent decades. In our previous work, we revealed its biosynthetic pathway through overexpression of the cluster-situated regulator gene *blnR* in Chinese herb fungus *Tolypocladium ophioglossoides*. However, information on the regulation of *blnR* is still largely unknown. In this study, we further investigated the regulation of balanol biosynthesis by BlnR through the analysis of affinity binding using EMSA and RNA-seq analysis. The results showed that BlnR positively regulates balanol biosynthesis through binding to all promoters of *bln* gene members, including its own promoter. Microscopic observation revealed *blnR* overexpression also affected spore development and hypha growth. Furthermore, RNA-seq analysis suggested that BlnR can regulate other genes outside of the balanol biosynthetic gene cluster, including those involved in conidiospore development. Finally, balanol production was further improved to 2187.39 mg/L using the optimized medium through statistical optimization based on response surface methodology.

## 1. Introduction

The fungal metabolite balanol was isolated as a potent ATP-competitive inhibitor of protein kinase C (PKC) from *Verticillium balanoides*, which was the same compound as that previously reported as azepinostatin from *Fusarium merisomides* and ophiocordin from *Tolypocladium ophioglossoides (*its synonymic name is *Cordyceps ophioglossoides)* [1,2,3,4]. It was shown to have inhibitory activity toward PKC isozymes in the nanomolar range, with better potency than the reported product staurosporine, as well as showing some activity against PKA. Protein kinase C (PKC) group can regulate the conformation and activity of target proteins by phosphorylating their serine or threonine residuals. PKC is the receptor for phorbol esters that promote tumor formation, playing crucial roles in cell proliferation and differentiation [5]. The upregulated activation of PKC has been related to a range of disease states, including central nervous system (CNS) diseases, cardiovascular disorders, diabetes, asthma and HIV infections [6,7]. The role of these enzymes in the development of cancer makes them an ideal target for screening their interesting inhibitory compounds. The specific molecular structure of balanol is similar to that of ATP, making it the ATP competitor binding to the PKC enzyme, thus inhibiting PKC activity.

*T. ophioglossoides* is a kind of parasitic mushroom of certain Elaphomyces, and it has been used to relieve postmenopausal syndrome in women in Chinese traditional medicine throughout history. Several active compounds have been identified from *T. ophioglossoides*, such as balanol, cordycepol and tyrosol [8,9,10]. Compared with the annotated gene clusters based on the genome sequence of *T. ophioglossoides*, many other gene clusters are still cryptic during laboratory culture. Although balanol has been produced through liquid fermentation, its yield was very low, even in large-scale fermentation. In recent decades, our group focused on the chemical synthesis of balanol, as well as its analogs, before elucidating f its biosynthetic pathway through overexpression of its cluster-situated regulator gene, *blnR* [10].

Several efficient strategies have been developed to activate cryptic gene clusters for the production of new compounds, such as microbial cocultivation [11,12], promoter engineering [13], ribosome engineering [14], epigenetic regulation [15,16] and transcription regulation by regulatory proteins, including global and pathway-specific regulators [17,18]. Transcription factors are usually involved in various important processes during microorganism growth and development via regulation of a series of target genes. Microbial genome sequence analysis reveals that transcription regulator-encoding genes are present within many individual gene clusters. Switching on the regulator gene in its active state is considered an important strategy to activate cryptic gene clusters to produce new secondary metabolites (SMs) via genetic engineering or improve the production of valuable products [19]. Compared with the global regulator, the pathway-specific activator gene is usually cluster-located, and its overexpression using a strong promoter or knockout is a simple and efficient strategy to improve the product of interest or activate the cryptic gene cluster. For example, the overexpression of the StrR family regulator in *Streptomyces* significantly increased ristomycin A content [20]. Moreover, SM production by microbes has been observed to vary with the composition of culture media and culture conditions [21]. Based on this concept, the approach of ‘one strain, many compounds’ (OSMAC) was developed by changing culture medium and has been widely used in many bacteria and fungi to efficiently mine novel SMs, including polyketides, non-ribosomal peptides and terpenes, in recent years [22,23,24]. Hence, changing culture medium is also a classic approach to activating cryptic gene clusters. In addition, cultivation parameters, including temperature, salinity and dissolved oxygen, are considered effective ways to trigger cryptic biosynthetic pathways in *Aspergillus ochraceus*.

With respect to the biosynthesis pathway of balanol, we previously characterized several key biosynthetic enzymes, including BlnJ, BlnF, BlnO and BlnP [10]. However, there is still a lack of regulatory information about the balanol biosynthesis. According to the annotation, there is only one regulatory gene, *blnR* was found within the gene cluster *bln*. BlnR is a Zn_2_Cys_6_ family regulator, but the role of BlnR in balanol production has not been investigated. Therefore, it is important to understand its regulatory function in balanol biosynthesis in order to improve its production. In this study, the regulation of balanol biosynthesis was first investigated by analyzing the affinity-binding ability between regulator BlnR and target genes using an EMSA experiment. Then, RNA-seq analysis was performed to investigate the effect of *blnR* overexpression on *T.*
*ophioglossoides*. Additionally, the culture conditions for balanol production by strain *blnR*OE were further optimized based on statistical experimentation through a one-factor experiment and response surface methodology (RSM). As a consequence, the cluster-situated regulator, BlnR, positively regulated balanol biosynthesis by binding to all the promoters of gene cluster members, as well as its own promoter. The transcriptomic data showed that BlnR is broadly involved in both the primary and secondary metabolites. The concentration of balanol in a 5 L fermenter tank was improved to 2187.39 mg/L from 700 mg/L through statistical optimization using the optimized medium.

## 2. Materials and Methods

### 2.1. Chemicals

Potato dextrose agar (PDA) and chlorimuron-ethyl were purchased from Sigma (St. Louis, MO, USA). Polypeptone, imidazole, Tris-HCl solution, kanamycin PMSF and IPTG were purchased from Sangon Biotechnology Incorporation (Shanghai, China). Yeast extract was purchased from Oxoid (England, UK). Sucrose (99.5%), MgSO_4_·7H_2_O (99%), KH_2_PO_4_ (99%), NaOH (96%), NaCl (99.5%) and trifluoroacetic acid (TFA, 99.5%) were purchased from China National Pharmaceutical Group Corporation (Shanghai, China). Acetonitrile (99.9%) was obtained from Astoon Chemical Technology Incorporation (Wilmington, DE, USA).

### 2.2. Strain, Media and Culture Conditions

*Escherichia coli* DH5α was used for routine DNA manipulation, and BL21 (DE3)/Origami B was used as a host for protein expression. *blnR* overexpression strain *T. ophioglossoides blnR*OE was used as the balanol-producing strain, which was routinely supplemented with 4 mg/L of chlorimuron-ethyl (Sigma, USA) if necessary, as previously described [10]. *E. coli* cells were cultured in LB broth at 37 °C and 200 rpm. *Agrobacterium tumefaciens* EHA105 was used to transfer DNA into *T. ophioglossoides* via T-DNA transformation, as described previously [25]. COB liquid medium (sucrose 30 g/L, polypeptone 5 g/L, yeast extract 5 g/L, MgSO_4_.7H_2_O 1 g/L, KH_2_PO_4_ 0.5 g/L, pH 5.5) was used as the seed culture and starter fermentation medium.

For balanol production, seed cultures were prepared by inoculating 2 × 10^5^ spores/mL of *blnR*OE strain in COB medium in 250 mL shake flasks with 80 mL medium and incubated at 26 °C and 160 rpm for 3–4 days. Then, seed cultures were transferred to the fermentation medium using 2.5% (*v*/*v*) inoculum. For each strain, a shake flask assay was carried out in triplicate parallel bottles. Batch fermentation was carried out in a 15 L jar fermenter at 26 °C (BLBIO-3GJ, China) containing 8 L of culture medium. The seed culture was prepared and inoculated into the fermenter jar at 2% (*v*/*v*). The pH of the medium was maintained at 4.9 via automatic addition of 2 M NaOH. The dissolved oxygen concentration was maintained at 20% air saturation by automatically increasing the agitation speed. Culture samples were periodically taken to analyze balanol concentration and CDWs. Both the batch and fed-batch fermentation experiments were carried out in triplicate.

### 2.3. Heterologous Expression of blnR in E. coli and Its Purification

Total RNA extracted from mycelia of the *blnR*OE strain was used as a template for first-strand cDNA by SuperScript™ IV reverse transcriptase (Invitrogen). A cDNA fragment encoding the whole length of *blnR* and its DNA binding domain (*blnR*^DBD^) of *T. ophioglossoides* was subcloned into the pET-32a vector (Novagen, Darmstadt, Germany) by infusion cloning technology (Vazyme, Nanjing, China). The BlnR^DBD^ and BlnR protein were produced in *E. coli* BL21 (DE3) cells by the addition of IPTG and grown overnight at 16 °C in a 1 L flask in the presence of 50 mg/L kanamycin. The pellet cells were collected by centrifugation, resuspended in 200 mL lysis buffer (20 mM Tris-HCl, 500 mM NaCl, 10 mM imidazole, 1 mM PMSF, pH 8.0) and disrupted by sonication. After centrifugation at 12,000× rpm at 4 °C for 5 min, the cleared cellular extract was adjusted to pH 8.0 and loaded on a Ni-agarose column (GE, Munich, Germany), which was previously equilibrated with 20 mM Tris-HCl, 500 mM NaCl and 20 mM imidazole (pH 8.0). The impurity protein was washed with 20 mM Tris-HCl, 500 mM NaCl and 100 mM imidazole (pH 8.0), and the BlnR^DBD^-containing fraction was eluted with elution buffer containing 500 mM imidazole (pH 8.0). The elution fraction was loaded on a desalting column with 3 kDa to be concentrated and redissolved in 20 mM Tris-HCl, 500 mM NaCl and 1 mM PMSF (pH 8.0).

### 2.4. Affinity Analysis by Electrophoretic Mobility Shift Assay (EMSA)

An electrophoretic mobility shift assay was performed to investigate the affinity of regulator BlnR with DNA fragments as described previously [26]. An FM-labeled DNA fragment was prepared by PCR amplification, directly using FM-labeled primers as EMSA probes. The purified PCR product was then employed as a template to generate the FM-labeled DNA probe using the corresponding FM-labeled primer, as shown in Appendix A. A volume of 1 μg salmon sperm DNA was used as a non-specific competitor in the binding mixture. The FM-labeled probes were detected with an LAS4000 machine (GE, Boston, MA, USA).

### 2.5. Phylogenetic Analysis

All the Zn_2_Cys_6_ transcription factors used for alignment were collected from the NCBI protein database by blast alignment. All the protein sequences were aligned using ClustalW in MEGA 7, and a phylogenetic tree was constructed with the maximum likelihood method based on the JTT matrix-based model. Protein domain architecture analysis was performed by conducting a search of the Conserved Domains Database (https://www.ncbi.nlm.nih.gov/cdd), as well as online analysis with Pfam (www.pfam.org).

### 2.6. mRNA-Seq Analysis and Differential Gene Expression Analysis

Total RNA was isolated from *T. ophioglossoides* mycelia grown in COB medium for 4 days at 26 °C by RNA extraction kits according to the manufacturer’s instruction (Takara, Japan). The genomic DNA in the RNA samples was digested by RNase-free DNase I (Takara, Japan). The first-strand cDNA was reverse-transcribed from total RNA with SuperScript™ IV reverse transcriptase (18090010, Invitrogen). qRT-PCR was performed using SYBR Premix Ex Taq II (Takara, Japan), and PCR procedures were performed at 95 °C for 5 min, 40 cycles of 95 °C for 15 s, 56 °C for 40 s and 72 °C for 20 s. The *Totef1* gene encoding housekeeping translational elongation factor was used as the internal control. The changes in target gene expression were quantified as 2^−ΔΔCt^ according to the manufacturer’s instructions (Takara, Japan). The primers used are listed in Appendix A.

For mRNA-seq analysis, 1 μg of total RNA was used as the input template for the RNA sample preparations. The sequencing libraries were generated using the NEBNext UltraTM RNA Library Prep Kit for Illumina (NEB, Ipswich, MA, USA) following the manufacturer’s recommendations, and index codes were added to attribute sequences to each sample. First-strand cDNA was synthesized using a random hexamer primer and M-MuLV reverse transcriptase. Second-strand cDNA synthesis was subsequently performed using DNA polymerase I and RNase H. Remaining overhangs were converted into blunt ends via exonuclease/polymerase activities. After adenylation of the 3′ ends of DNA fragments, NEBNext Adaptors with hairpin loop structure were ligated to prepare for hybridization. The library fragments were purified with an AMPure XP system (Beckman Coulter, Beverly, MA, USA) to choose cDNA fragments with a preferential length of 240 bp. Then, USER enzyme (NEB, Ipswich, MA, USA) was used with adaptor-ligated cDNA at 37 °C for 15 min, followed by 5 min at 95 °C before PCR. PCR was performed using Phusion high-fidelity DNA polymerase, universal PCR primers and index (X) Primer. Finally, PCR products were purified (AMPure XP system), and their quality was evaluated by the Agilent Bioanalyzer 2100 system. Clustering of the index-coded samples was performed on a cBot cluster-generation system using a TruSeq PE v4-cBot-HS cluster kit (Illumina) according to the manufacturer’s instructions. After cluster generation, the library preparations were sequenced on an Illumina NovaSeq 6000 platform, and paired-end reads were generated. The reference genome of *T. ophioglossoides* was predefined for analysis and mapping of RNA-seq reads with an HISAT2 system.

Differential expression analysis of genes between two samples was performed using the EdgeR bioconductor package and a dispersion parameter of 0.1. EdgeR provided statistical routines to determine differential expression in digital gene expression data using a model based on the negative binomial distribution. The resulting *P* values were adjusted using the Benjamini–Hochberg approach to control the false discovery rate. Genes with an adjusted *p*-value < 0.05, as determined by EdgeR, were classified as differentially expressed.

### 2.7. Optimization of Medium Components for Balanol Production by Response Surface Methodology (RSM)

The Box−Behnken statistical method was used for the optimization of the medium components. Critical parameters were observed, namely sucrose, polypeptone and initial pH. Design-Expert^®^ 10.0.0 software was employed for experimental design and analysis. A total of 12 run experiments were tested based on the design matrix with three center points to minimize the experimental error. A model was generated based on the response values of balanol production, and statistical significance was tested by analysis of variance (ANOVA). The predicted combination of medium components for maximum balanol production was further validated experimentally.

The optimal medium was statistically optimized through response surface methodology in the *blnR*OE strain. The effect of one factor on balanol production was first examined, and then PB design was applied to determine the significant components according to the balanol production, including 12 run experiments and 6 variables, including sucrose, polypeptone, yeast extract, KH_2_PO_4_, MgSO_4_ 7H_2_O and pH (Table 1). Based on the results of the PB, the significant factors were further optimized based on the RSM coupled with BB design using Design-Expert software to determine the final optimal fermentation medium. The quality of fit of the second-order polynomial model equation was determined via a coefficient of determination (R^2^) and the adjusted R^2^. ANOVA (analysis of variance) was used for graphical analyses to estimate the interaction between the component variables and balanol production. The components in the culture media that showed confidence levels >95% were considered to exhibit significant responses to balanol production.

### 2.8. Analysis of Balanol Production by HPLC

The concentration of balanol was determined according to the method described by He et al. (2018) [10]. Culture broth was sampled for analysis of balanol by HPLC using a reverse-phase C18 column (Agilent Eclipse Plus C18, 4.6 × 250 mm, 5 μm) (1260 Infinity, Agilent Technologies, Santa Clara, CA, USA). Chromatographic conditions were composed of solvent A and solvent B. Solvent A comprised water with 0.001 M trifluoroacetic acid (TFA), and solvent B comprised acetonitrile-0.001 M TFA; the solvent gradient was 5% B in the first 5 min and increased to 58% at 35 min and to 95% B at 36 min, followed by 4 min with 95% B, with a flow rate of 1 mL/min and UV detection at 254 nm. The structure-identified balanol was used as the standard control. Through analysis, the peak area of balanol with different concentrations was determined by HPLC, and the standard curve of balanol concentration was established (Figure 1). The concentration of balanol production in culture broth was determined with the regression equation from the standard curve: Y = 31.764X − 203.51 (R^2^ = 0.9993), where Y indicates the concentration of balanol (mg/L), and X is the peak area of balanol.

## 3. Results

### 3.1. Characterization of Regulator BlnR within the Gene Cluster bln in T. ophioglossoides

The gene *blnR* was found as an orphan regulatory gene situated within the *bln* cluster in *T. ophioglossoides*, which connects the PKS and NRPS part of gene cluster *bln* containing 18 genes, with a length of 79 kb. The *blnR* gene is 1443 bp long without intron and encodes a protein BlnR containing 380 amino acids. Through bioinformatic analysis, BlnR was determined to be a putative transcription regulator featuring a typical N-terminal GAL4-type Zn_2_Cys_6_ DNA-binding domain and a C-terminal AflR domain (Figure 2A). Like other GAL4 domains among the aligned proteins, the six cysteine residues were conserved in the putative GAL4 domain of BlnR (Figure 2B). The transcription factor AflR contains a GAL4-type binuclear zinc finger cluster domain, CX2CX6CX6CX2CX6CX2, which plays a key role in aflatoxin biosynthesis, especially in *Aspergillus* taxa [27,28]. The six-cysteine (Zn_2_Cys_6_) binuclear cluster DNA binding domain was first characterized in the GAL4 protein of *Saccharomyces cerevisiae*. To date, this domain-containing protein has been identified exclusively in the fungus kingdom [29,30]. Phylogenic analysis showed that BlnR belongs to a separate clade from *Aspergillus* AflR regulator protein, showing it to be different from the AflR of Aspergillus with a different function (Figure 2C).

### 3.2. BlnR Positively Regulates Balanol Biosynthesis by Binding All the Promoters of bln Gene Members

Usually, a typical transcriptional activator consists of a DNA-binding domain (DBD), which is responsible for promoter recognition in order to regulate gene transcription. Nothing is known yet about the regulation of balanol biosynthesis, although we have elucidated its biosynthetic pathway in *T. ophioglossoides*. Therefore, we first attempted to heterologously express the whole length of BlnR in the *E. coli* system. Unfortunately, both BlnR^360aa^ and BlnR^180aa^ were expressed as inclusion bodies in *E. coli*, even with the help of solubilizing tag protein GST. Therefore, we attempted to only express its N-terminal GAL4 domain with 90 aa in *E. coli*. As shown in Figure 3B, the fusion GST-BlnR^90aa^ was expressed successfully as a soluble protein. After purification by Ni-affinity agarose, GST-BlnR^90aa^ was used for the DNA-binding experiment. Using FM-labelled primer pairs, 12 promoter fragments of the *bln* gene cluster were amplified for EMSA assay to examine the affinity of BlnR protein to their promoters of the *bln* gene members. As the results revealed, BlnR^90aa^ showed a strong affinity with all the tested promoters, as well as its own promoter (Figure 3C). These results are consistent with the upregulation of their transcription level in the *blnR*OE strain as compared with the wild type [10]. It is reasonable to speculate that there a conserved binding site exists in the promoter region of all *bln* gene members. Therefore, we carried out motif investigation to mine DNA-binding motifs by multiple expectation maximizations for motif elicitation (MEME) (https://meme-suite.org). It was demonstrated that there is a conserved motif in all promoter regions with GAGCCAAT (Figure 3D). 

### 3.3. BlnR Is a Positive Regulator toward Balanol Biosynthesis in T. ophioglossoides

In *T. ophioglossoides*, there are another 34 gene clusters aside from *bln* according to analysis by AntiSMASH (https://fungismash.secondarymetabolites.org/, accessed on 18 April 2022). We showed that the overexpression of the *blnR* gene significantly upregulated the transcription level of all the *bln* member genes and activated the biosynthesis of balanol. The results suggest that BlnR has a positive regulatory effect on balanol biosynthesis. The *blnR*OE strain remained stable for balanol production after 10 generations grown on a PDA plate without selective pressure. Furthermore, we examined the effect of *blnR* overexpression on the transcriptional level of all other gene clusters by comparing their core gene in the wild-type and *blnR*OE strain. The results demonstrate that *blnR* overexpression exclusively improved the core gene expression of the balanol gene cluster and insignificantly ffected the expression of other gene clusters (Figure 4A). The metabolite profiles by HPLC analysis also exhibite that no other compounds were producedexcept balanol and its intermediates (Figure 4B).

### 3.4. BlnR Is Involved in the Crosstalk between Primary and Secondary Metabolism

When grown on a PDA plate, we found *blnR* overexpression changed its morphologic phenotype. In addition to the production of light-yellow pigment of stable balanol, the *blnR*OE strain also showed slowed growth as compared with the wild-type strain (Figure 5A,B). Through scanning electron microscopy, the number of conidiospores was found to be reduced significantly, and its slimly hyphae elongated and became curly in the *blnR*OE strain, whereas the grown hyphae were linearly elongated with numerous conidia visible in the wild-type strain with typical morphology (Figure 5C). Therefore, we further analyzed the key central regulators involved in conidiation development, such as *brlA* and *abaA,* as well as their upstream genes, *fluG*, *flbC* and *flbD*. *abaA*, *fluG* and *flbD* were upregulated, whereas *brlA* and *flbC* were slightly downregulated in the *blnR*OE strain (Figure 5D).

We further investigated the differential expression pattern between *blnR*OE and the wild-type strain through mRNA-seq analysis. As shown in Figure 6, *blnR* overexpression upregulated the transcription level of 498 genes and downregulated the transcription level of 503 genes, whereas 8316 genes maintained their expression at a regular level. RNA-seq data analysis revealed the differential expression of numerous genes belonging to various pathways of primary or secondary metabolism. Pathways of primary metabolism include starch and sucrose metabolism, fatty acid metabolite, TCA cycle, lysine biosynthesis, ether lipid metabolism and aromatic amino acid (phe, tyr, trp) biosynthesis (Table 2). Among them, the expression of most genes in starch and sucrose metabolism, lysine biosynthesis and aromatic amino acid biosynthesis were significantly upregulated in the *blnR*OE strain, whereas genes in fatty acid metabolite and TCA cycle were up- or downregulated. The genes expression in the pathway of spore development was significantly downregulated, except for the *bln* gene cluster, which was activated strongly in *blnR*OE as compared with the wild-type strain. Our results show that as a cluster-situated regulator, the overexpression of the *blnR* gene is involved in the control of secondary metabolism, as well as primary metabolism, possibly by manipulating the distant genes.

### 3.5. High Production of Balanol at the Gram Level through Medium Optimization via Response Surface Methodology (RSM)

In order to further enhance balanol production, the effect of condition parameters, including inoculum dosage, pH, carbon source and nitrogen source on balanol production was investigated through PB experiments. As shown in Figure 7, sucrose, which varied from 30 g/L to 120 g/L, showed a significant effect on balanol production and cell growth, whereas 105 g/L of sucrose was determined to be the optimal concentration (Figure 7A), and 15 g/L polypeptone as the nitrogen source (Figure 7B) or 10 g/L of the yeast extract (Figure 7C) was the best concentration with the maximum titer of balanol. The effect of the combined nitrogen source further demonstrates that balanol production reached the maximum at 5 g/L yeast extract and 10 g/L of polypeptone (Figure 7D).

The significant factors affecting production as observed in OFAT, namely (A) sucrose, (B) polypeptone and (C) yeast extract, were studied for their optimum combination using RSM and adhering to the PB and BB design matrix. A total of 12 run experiments were performed according to the PB matrix to investigate the significant components in balanol production (Table 1). Regression statistics were performed to examine the model feasibility. As shown in Table 3, a model with a *p*-value lower than 0.05 was considered significant, and sucrose, polypeptone and initial pH were determined to be the main significant components affecting balanol production. Therefore, a 17-full factorial BB experimental design was further implemented to determine their optimal values based on RSM. The experimental design matrices that included all the variables and balanol titers are shown in Appendix A. The resulting fermentation titers of balanol were used to fit a quadratic model using regression analysis, yielding the following response equation to predict balanol production in terms of coded variables:Y = 1814.86 − 21.55A − 103.12B + 332.95C + 107.44AB − 33.82AC + 54.17BC − 585.06A^2^ − 365.13B^2^ − 434.71C^2^
where Y indicates the balanol production (mg/L); and A, B and C are sucrose, polypeptone and pH, respectively. Moreover, we used this statistical model to evaluate the relationship between different variables and their interactive effects on balanol production, as summarized in Table 4. Regression analysis of BB design showed that the model’s *F* value was 32.35, and the model’s *p*-value was lower than 0.0001 with statistical significance, suggesting that this model is fit well to describe the response of balanol production of these variable components.

According to the BB model, the individual and interactive effects of the independent component on the response of balanol production are further shown by contour plots to predict the response surface; the balanol titer was a function of two tested variables, whereas the other independent variable was kept constant at the zero level. Among these components, the interactive effect between two components on balanol production is given in Figure 8. Balanol titers initially increased with concentrations of both polypeptone and pH, then declined after reaching the maximum point in each variable, showing that there was a curve relationship between the independent variables and balanol production (Figure 8A). A similar interaction was independently observed between sucrose and polypeptones, sucrose and pH (Figure 8B,C). However, the *p*-values for the interactive term of these pairs of variables were higher than 0.05, suggesting statistically insignificant interactions between these three components. It can be inferred that the central values of these three variants subjected to BB design were close to their optimal values. According to the RSM model, the optimal values for the maximum production of balanol were 100 g/L sucrose, 13.6 g/L polypeptone and an initial pH of 4.9.

### 3.6. Batch Fermentation for Balanol Production in a Scaled-Up 15 L Tank

Using the optimal medium compositions, including 100 g/L sucrose, 13.6 g/L polypeptone, 5 g/L yeast extract, 0.6 g/L KH_2_PO_4_, 1.0 g/L MgSO_4_·7H_2_O, initial pH and 2% inoculum volume, we carried out a scaled-up batch fermentation with the *blnR*OE strain in a 15 L fermenter. As can be seen from the time course of the fermentative profile shown in Figure 9A, the maximum concentration of balanol reached 2187.39 mg/L after culture for 10 days, and the cell biomass was found to be the highest after 8 d growth. The results matched well with the predicted value of the developed statistical model, suggesting that this model truly reflects the effect of medium components on balanol production. It provides a feasible practical attempt for large-scale industry production. We also analyzed the effect of optimized medium on the expression of gene members of the *bln* gene cluster (Figure 9B). The results show that their expression did not differ significantly when cultured in the optimized medium.

## 4. Discussion

Fungi micro-organisms are a rich source for producing novel compounds with potential bioactivity. With the development of sequencing technology, bioinformatic analysis has revealed that a far larger number of gene clusters is cryptic in the genomic sequence. Mining the genomic source to discover novel compounds is attractive. The secondary metabolism is a complex network regulated by global regulators, such as LaeA and velvet [31,32]. In the developed strategies, transcription regulation has proven to be a simple and feasible method to activate cryptic biosynthetic gene clusters or improve the production of compounds of interest. It is known that transcription factors can directly regulate the transcription of their target genes by binding to promoters, leading to an improvement or reduction in the production of target compounds. Recently, a targeted and high-throughput activation of silent gene clusters using transcription factor decoys was applied in Streptomycetes [33]. In this study, we identified a cluster-situated Zn_2_Cys_6_ family regulator, BlnR, which plays a positive role in regulation of balanol biosynthesis by binding to all promoters of gene cluster members, as well as its own promoter, within the *bln* (Figure 3C). Overexpression of the *blnR* gene significantly activated the transcription levels of all gene members within the *bln* gene cluster.

Additionally, mRNA-seq analysis exhibited that the overexpression of *blnR* led to 1001 differentially expressedgenes, which are involved in different pathways, including primary and secondary metabolism, in the *blnR*OE strain (Table 2). For primary metabolism, such as starch and sucrose metabolism, fatty acid metabolism and TCA cycle, there are up-regulated genes and downregulated genes. Aromatic amino acid biosynthesis and lysin-biosynthesis-related genes were upregulated significantly, which was reasonable because lysine and phenylalanine are the substrate of NRPS for balanol biosynthesis within the gene *bln* cluster. All these data indicate that BlnR is involved in the coordination of secondary metabolism and primary metabolism to promote balanol biosynthesis.

We also found that *blnR* overexpression resulted in changes in its physiological phenotype in the *blnR*OE strain. Conidiospore development was blocked, with a reducing number, and the hyphae grew curly with retarded growth (Figure 5). Conidiation is considered the most common asexual reproductive mode for many filamentous fungi, and its developmental mechanisms have been characterized in *A. nidulans* and *Neurospora crassa* [34,35]. Transcription analysis showed that regulators involved in spore development, such as AbaA, FluG and FlbD, were upregulated. RNA-seq data also revealed that outer spore wall protein RRT8 and spore development regulator vosA were downregulated significantly in the *blnR*OE strain (Figure 5 and Table 2). These data suggest that the *blnR* gene is also involved in spore development with the slowed growth of the hyphae.

Previous studies have shown that AflR can regulate the expression of genes outside of the aflatoxin biosynthetic cluster under conditions conducive to aflatoxin production in *A. parasiticus* and *A. flavus* [36,37], suggesting that AflR may have a broad function and regulate other distant genes. Consistent with these results, our study also showed that BlnR can regulate the genes within the balanol biosynthetic cluster, as well as other distant genes involved in many other metabolic pathways (Figure 6 and Table 2), which are directly regulated by BlnR and will be crucial for further studies.

Genetic manipulation was proven to be an efficient technique to activate and prove the biosynthesis of secondary metabolites. Meanwhile, the OSMAS cultivation-based technique can also powerfully activate or increase SM production by changing the culture conditions, including the proper ratio of carbon to nitrogen, metal ions, pH and temperature. Many novel compounds were found through this simple strategy. In large-scale industrial production, the optimization of culture conditions is used as a feasible way to increase the production of target products in order to reduce economic costs. In this study, we carried out medium optimization using RSM-based statistics and further improved balanol production by 3.12-fold to 2187.39 mg/L in a 15 L fermenter by increasing the transcription level of gene members of the *bln* gene cluster (Figure 9A). The optimized medium did not further enhance the expression of all *bln* gene members, revealing that changing culture conditions possibly led to the alteration of enzyme activity or other metabolite pathways.

## 5. Conclusions

In conclusion, the cluster-situated Zn_2_Cys_6_-family regulator, BlnR, has a positive and specific regulatory effect on balanol biosynthesis in *T. ophioglossoides*. BlnR was found to regulate the other genes outside of the balanol biosynthetic gene cluster, including the primary and secondary metabolite pathways. In addition, BlnR was also found to be involved in the development of asexual conidiospores and mycelium growth. Furthermore, statistical methods based on RSM were used to determine optimal medium compositions with a maximum titer of balanol in the *blnR*OE strain. Using these optimized components, the highest concentration of balanol was determined to be 2187.39 mg/L after 10 d cultivation in a 15 L batch tank.

## Figures and Tables

**Figure 1 jof-08-00510-f001:**
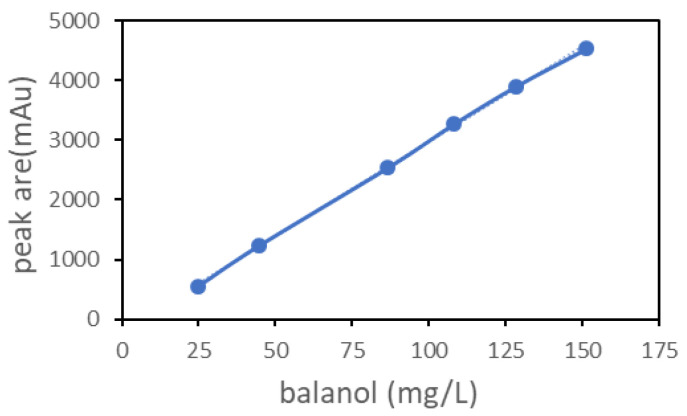
Standard curve of the balanol. Balanol was prepared in a solution from 25 mg/L to 180 mg/L for HPLC analysis. The peak area of balanol was used to establish the standard curve of its concentration. In the regression equation, Y = 31.764x − 203.51 (R^2^ = 0.9993), Y indicates the concentration of balanol, and X indicates the peak area of balanol.

**Figure 2 jof-08-00510-f002:**
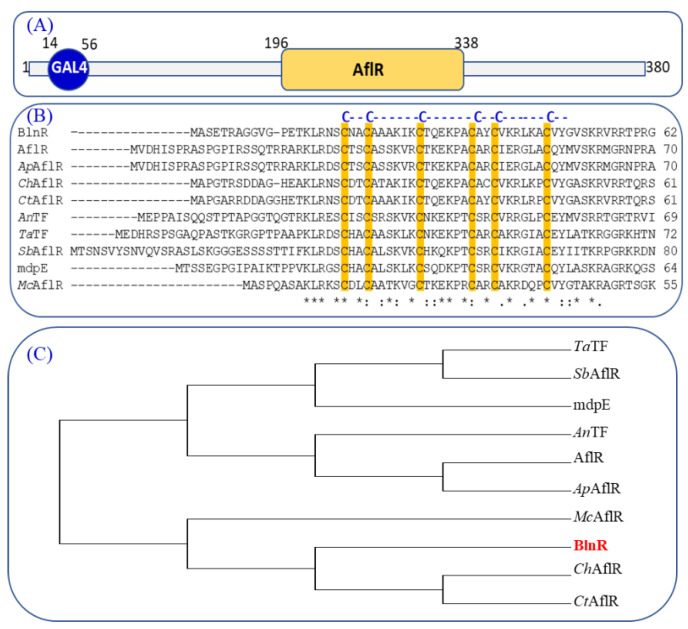
Characterization of regulator BlnR and its phylogenic analysis. (**A**) Domain characterization of regulator BlnR. (**B**) Alignment analysis of the conserved cysteine amino acids. (**C**) Evolutionary phylogenetic analysis by maximum likelihood method using MEGA 7.0 software. The evolutionary history was inferred using the maximum likelihood method based on the JTT matrix-based model: *Ta*TF (accession no. XP_040731924.1), *Sb*AflR (accession no. ESZ98975.1), mdpE (accession no. AN0148), *An*TF(accession no. AAC49195), AflR (accession no. P43651.3), *Ap*AflR (accession no. AAM02999.1), *Mc*AflR (accession no. XP_002844737.1), *Ch*AlfR (accession no. XP_018155792.1), *Ct*AflR (Accession no. TKW57950.1).

**Figure 3 jof-08-00510-f003:**
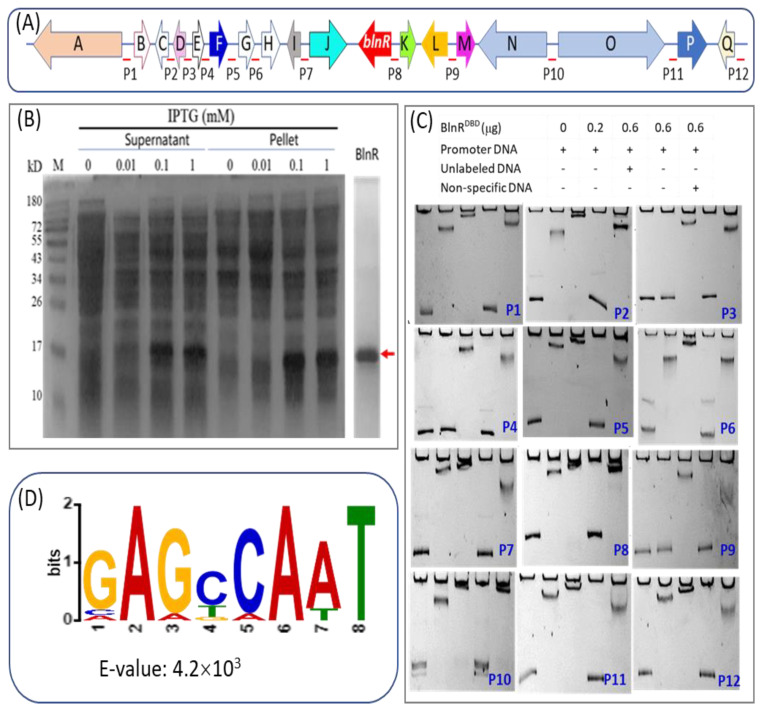
Affinity binding analysis of BlnR with promoter DNA through EMSA experiment. (**A**) Schematic diagram of promoter design in the *bln* gene cluster. (**B**) Heterologous expression of BlnR in *E. coli.* M: protein marker. (**C**) Affinity binding analysis of BlnR to gene promoters within *bln* by EMSA. (**D**) The conserved binding motif of BlnR was predicted by MEME analysis.

**Figure 4 jof-08-00510-f004:**
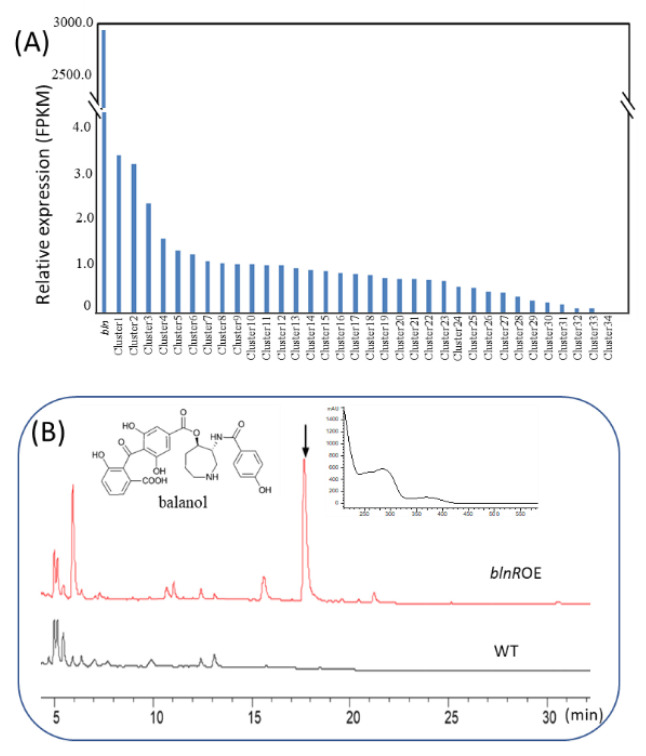
BlnR is a positive regulator of balanol biosynthesis. (**A**) Effect of *blnR* overexpression on the transcriptional level of other gene clusters. (**B**) Metabolite profile of wild-type and *blnR*OE strains after cultivation in COB medium for 10 days.

**Figure 5 jof-08-00510-f005:**
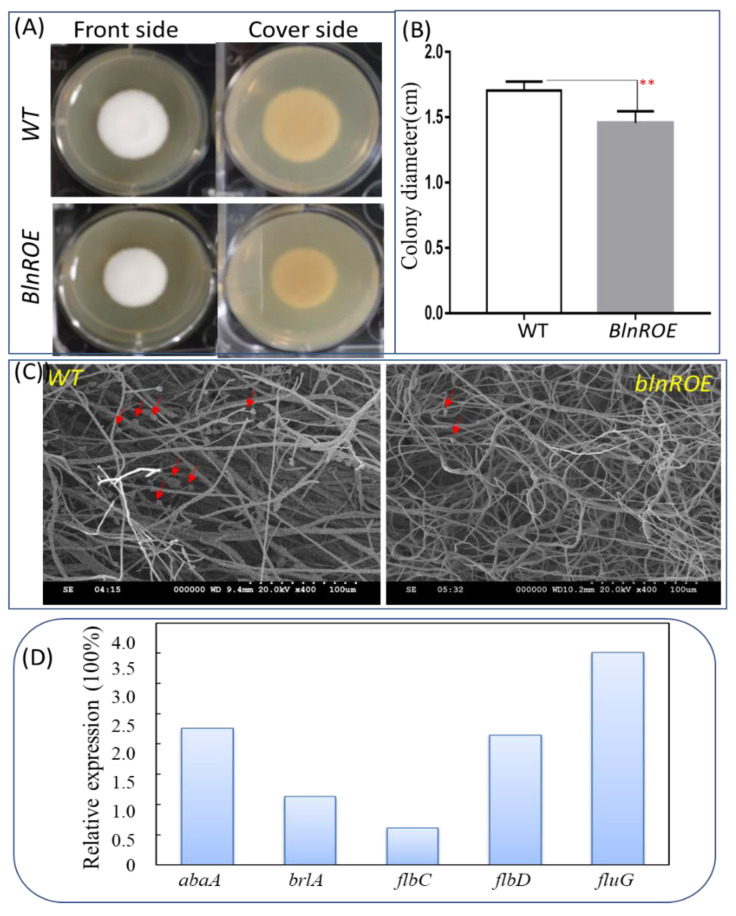
Effects of BlnR on the development of asexual conidiospore and filamentous growth. (**A**) A total of 10^3^ spores were spotted on a PDA plate for growth at 26 °C for 7 days. (**B**) Comparison of colony size between the wild-type and *blnR*OVE (n = 10) strains. “**” indicates the significant difference with the *p*-value < 0.01. (**C**) Microscopic observation of hyphae and spores. The red arrows indicate spores. (**D**) qRT-PCR analysis of spore-development-related regulators (n = 3).

**Figure 6 jof-08-00510-f006:**
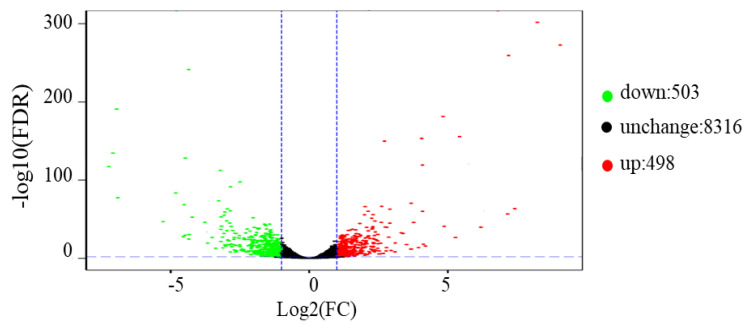
Differential expression analysis between *blnR*OE and the wild-type strain.

**Figure 7 jof-08-00510-f007:**
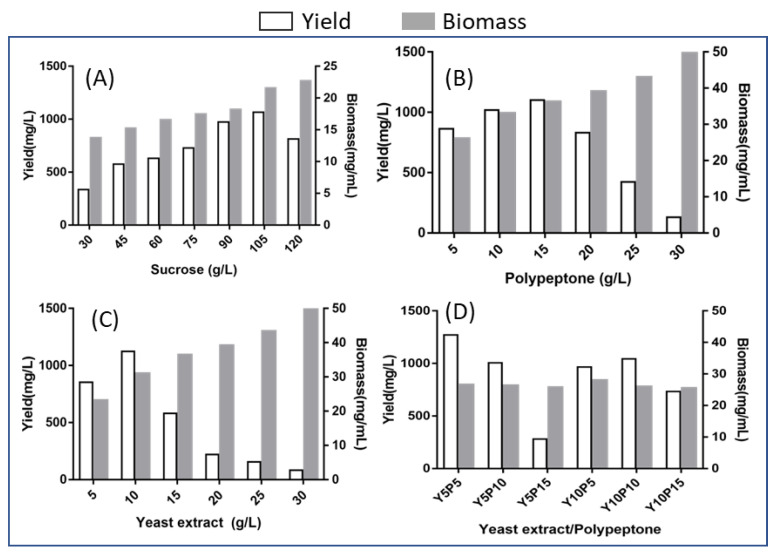
Effect of nutritional factor on balanol production including sucrose (**A**), polypeptone (**B**), yeast extract (**C**) and combined yeast extract and polypeptone (**D**).

**Figure 8 jof-08-00510-f008:**
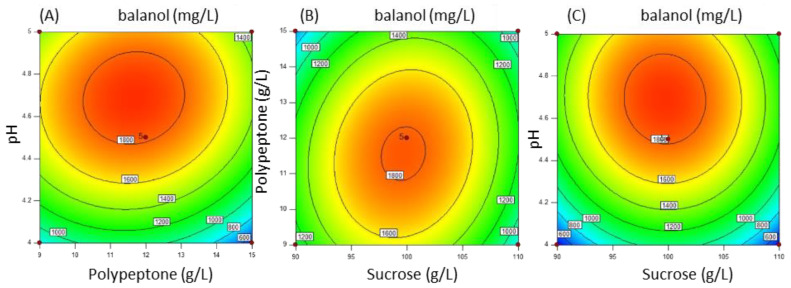
Contour plots for response interaction between two variables. (**A**) Interaction effect of pH and polypeptone on balanol production; (**B**) Interaction effect of sucrose and polypeptone on balanol production; (**C**) Interaction effect of pH and sucrose on balanol production.

**Figure 9 jof-08-00510-f009:**
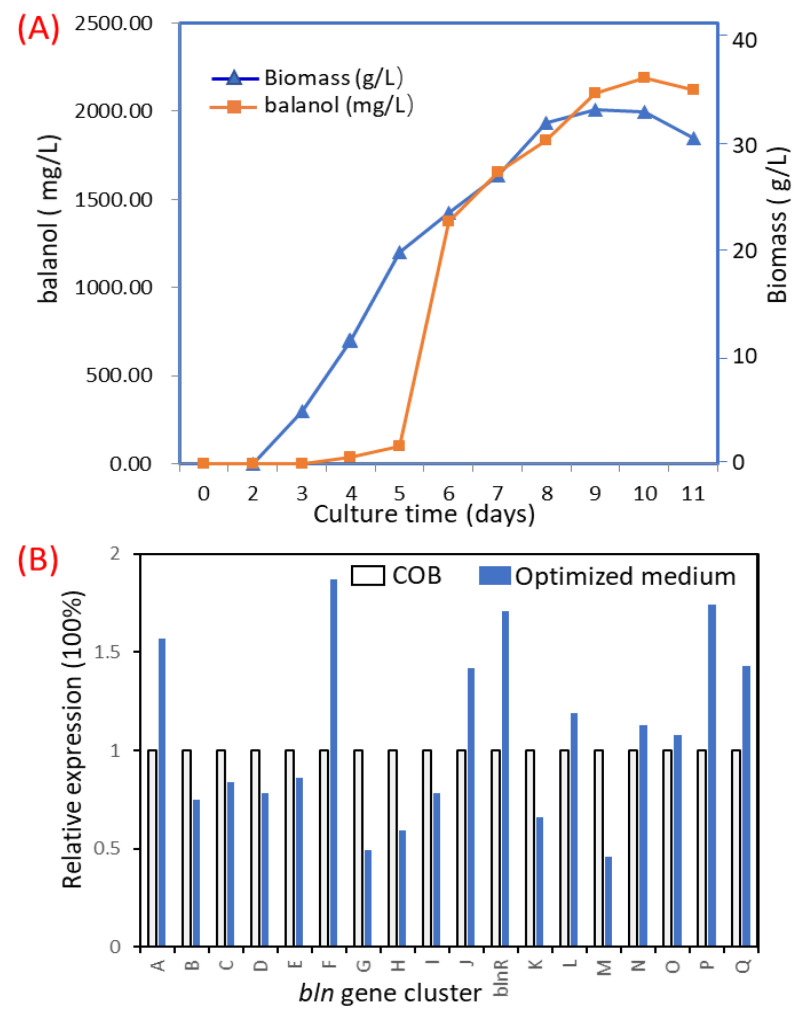
Time profile of balanol production by *blnR*OE strain using the optimized medium (**A**) and analysis of the expression level of the *bln* gene cluster (**B**). The *blnR*OE strain was cultured in a 15 L tank at 26 °C for 11 days with 8 L of medium. The culture broth was sampled for analysis of cell growth by determination of dried weight and balanol production every two days. The dissolved oxygen was maintained at 30% through cultivation. For RNA extraction, 4-day mycelium was used to analyze the expression level.

**Table 1 jof-08-00510-t001:** PB experimental design and observed balanol production.

Run	A	B	C	D	E	F	G	Balanol (mg/L)
1	1	−1	−1	−1	1	−1	1	1218.48
2	1	−1	1	1	−1	1	1	586.54
3	−1	−1	−1	−1	−1	−1	−1	832.88
4	−1	−1	−1	1	−1	1	1	633.36
5	−1	1	−1	1	1	−1	1	400.22
6	−1	1	1	1	−1	−1	−1	318.68
7	1	1	−1	−1	−1	1	−1	638.92
8	1	1	1	−1	−1	−1	1	1197.94
9	1	1	−1	1	1	1	−1	579.02
10	−1	−1	1	−1	1	1	−1	405.68
11	1	−1	1	1	1	−1	−1	1633.4
12	−1	1	1	−1	1	1	1	143.46

A: sucrose (g/L); B: polypeptone; C: yeast extract; D: KH_2_PO_4_; E: MgSO_4_ 7H_2_O; F: pH, G: inoculum volume (*v*/*v*).

**Table 2 jof-08-00510-t002:** List of genes up- or down regulated in *blnR*OE as compared with the wild-type strain based on RNA-seq analysis.

Gene ID	Protein Name	log2FC
** Starch and sucrose metabolism**		
g1571	putative betaglucosidase I	1.797
g2561	putative sucrose utilization protein SUC1	1.574
g2754	Hexokinase-1	1.715
g6339	Alphaamylase A type-3	1.338
g6369	Alphaglucosidase	1.387
g6411	Endoglucanase EG-II	−1.405
g6822	Probable betaglucosidase A	1.301
g7696	alphatrehalose-phosphate synthase	1.388
g857	endo-1,3-betaglucosidase eglC	1.360
g9295	Glucose-6-phosphate isomerase	−1.422
** Fatty acid metabolism**		
g507	Cytochrome P450	−2.278
g672	putative aldehyde dehydrogenase	−1.201
g2909	3-ketoacyl-CoA thiolase	−1.931
g3641	Acetyl-CoA acetyltransferase	−1.198
g3915	Enoyl-CoA isomerase/hydratase	−1.153
g5070	Enoyl-(Acyl carrier protein) reductase	2.751
g5094	Short-chain-type dehydrogenase/reductase	11.281
g5215	Acyl-CoA dehydrogenase family member 10	1.212
g6087	Phosphotransferase	−1.056
g6743	Aldehyde dehydrogenase	1.447
g7131	Acetoacetyl-CoA reductase	1.059
g7814	Isotrichodermin C-15 hydroxylase	−1.498
g9557	Short/branched-chain-specific acyl-CoA dehydrogenase	−1.433
** Citrate cycle (TCA cycle)**		
g3392	2-methylcitrate synthase	−1.435
g8077	Succinyl-CoA ligase	1.935
g9576	putative succinate dehydrogenase	−1.168
** Lysine biosynthesis**		
g3241	Homoaconitase, mitochondrial	1.077
g6260	Homocitrate synthase, mitochondrial	1.129
** Phenylalanine, tyrosine and tryptophan biosynthesis**		
g6745	Fungal-specific transcription factor	2.344
** MAPK signaling pathway**		
g7635	Catalase	−1.13221
g9623	Catalase	−3.62842
** Spore development**		
g6286	Spore development regulator vosA	−1.22895
g655	Outer spore wall protein RRT8	−0.94555

**Table 3 jof-08-00510-t003:** Signification analysis of the PB experiments on balanol production.

Source	SS	DF	MS	*F*-Value	*p*-Value
Model	4.336 × 10^5^	7	6.821 × 10^5^	6.82	0.0413 *
A	2.028 × 10^5^	1	2.028 × 10^5^	20.27	0.0108 *
B	8.603 × 10^4^	1	8.603 × 10^5^	8.60	0.04727
C	4.005 × 10^5^	1	6.15	6.145 × 10^−4^	0.9814
D	1.706 × 10^3^	1	1.706 × 10^3^	0.12	0.7009
E	5.495 × 10^4^	1	6.159 × 10^2^	0.044	0.8163
F	6.16 × 10^2^	1	1.424 × 10^5^	10.18	0.0188 *
G	1.424 × 10^5^	1	4.390 × 10^5^	1.55	0.2677
Residual	4.003 × 10^5^	4	1.001 × 10^5^		
Cor Total	5.175 × 10^5^	11			

* indicates that the effect of the variable is significant.

**Table 4 jof-08-00510-t004:** Variance analysis of the binary regression equation.

Source	SS	DF	MS	*F*-Value	Pr > F
Model	4.148 × 10^6^	9	4.609 × 10^5^	32.35	<0.0001 **
A	3.715 × 10^4^	1	3.715 × 10^4^	0.26	0.6253
B	8.506 × 10^5^	1	8.506 × 10^5^	5.97	0.0445 *
C	8.869 × 10^5^	1	8.869 × 10^5^	62.25	<0.0001 **
AB	4.618 × 10^5^	1	4.618 × 10^5^	3.24	0.1148
AC	4.575 × 10^3^	1	4.575 × 10^3^	0.32	0.5886
BC	1.174 × 10^5^	1	1.174 × 10^5^	0.82	0.3942
A2	1.441 × 10^6^	1	1.441 × 10^6^	101.15	<0.0001 **
B2	5.614 × 10^5^	1	5.614 × 10^5^	39.40	0.0004 **
C2	7.957 × 10^5^	1	7.957 × 10^5^	55.85	0.0001 **
Lack of Fit	2.485 × 10^5^	3	8.283 × 10^3^	0.44	0.7355
Pure Error	7.488 × 10^5^	4	1.872 × 10^5^		
Cor Total	4.248 × 10^6^	16			

* and ** indicate that the effect of the variable is significant and more significant, respectively.

## Data Availability

The data presented in this study are available upon request from the corresponding author. The RNA-seq data are not publicly available because other data from these whole-genome transcriptomes are being used for other analyses to be published independently of this one.

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
