# Peer review of "Gram-Level Production of Balanol through Regulatory Pathway and Medium Optimization in Herb Fungus Tolypocladium ophioglossoides"

_jof, 2022, doi:10.3390/jof8050510_

Round 1

Reviewer 1 Report

The manuscript in reference describes an interesting study on the production of balanol, a bioactive metabolite produced by T. ophioglossoides. The manuscript has relevant information and results that will be interesting for readers. However, the authors should address some issues prior to further consideration.

  1. Detailed scrutiny should be performed throughout the manuscript to revise/correct some grammar and stylistic issues since some passages and sentences are difficult to be followed. An editing service is then recommended.
  2. Revise in detail the M&M section. Some experimental details are missing to ensure outcome reproducibility. For instance, the brand, model, and grade of reagents, solvents, materials, and instruments must be provided.
  3. Line 98: milliliters must be abbreviated as “mL” instead of “ml”. be consistent throughout the manuscript.
  4. Line 170: which reference genome/transcriptome was used for read alignment? Was it assembled de novo? This information must be clearly informed.
  5. Line 207: indicate if the balanol concentration was measured by an internal or external standard. If an external standard was used, the standard curve details and the units to express the results (e.g., milligrams balanol per liter liquid culture) should be provided.
  6. Revise the text format since some paragraphs have different line spacing (e.g., lines 385-429).
  7. The discussion section can be improved since it is laconically developed. It can be even reorganized, possibly by subheadings. The manuscript has numerous significant results that have not been adequately discussed. In fact, section 3 is labeled as “Results and discussion” (line 208), but an additional section 4 as Discussion is included (line 384). Revise this apparent mistake.
  8. Lines 416-418: why speculate? Any reference to support this statement?
  9. Conclusions are pretty general and even disordered so that authors can include more specific findings from a mechanistic point of view.

Author Response

Response to the comments one by one as the following:

 1.Detailed scrutiny should be performed throughout the manuscript to revise/correct some grammar and stylistic issues since some passages and sentences are difficult to be followed. An editing service is then recommended.

Response: We have performed English editing service as the recommended (NO.43746) as well as checking the whole manuscript carefully.

2. Revise in detail the M&M section. Some experimental details are missing to ensure outcome reproducibility. For instance, the brand, model, and grade of reagents, solvents, materials, and instruments must be provided.

Response:

We have added the information of all Chemicals used in this study as shown in the part 2.1 Chemicals (Line 92-Line99). All the used instruments were also listed in the manuscript.

3. Line 98: milliliters must be abbreviated as “mL” instead of “ml”. be consistent throughout the manuscript.

Response:Thank you for your careful reading. We have abbreviated all milliliters as “mL” instead of “ml”.

4. Line 170: which reference genome/transcriptome was used for read alignment? Was it assembled de novo? This information must be clearly informed.

Response:We used the reference genome of T. ophioglossoides for read alignment directly as shown in Line 185-Line 187.

5. Line 207: indicate if the balanol concentration was measured by an internal or external standard. If an external standard was used, the standard curve details and the units to express the results (e.g., milligrams balanol per liter liquid culture) should be provided.

Response: The purified and identified balanol was used as external standard. The standard curve was prepared to determine the concentration of balanol as shown in Figure 1. All the balanol production in the manuscript was shown as milligrams per liter liquid culture.

6. Revise the text format since some paragraphs have different line spacing (e.g., lines 385-429).

Response: Thank you for your carefully reading. We have uniformed the line spacing.

7. The discussion section can be improved since it is laconically developed. It can be even reorganized, possibly by subheadings. The manuscript has numerous significant results that have not been adequately discussed. In fact, section 3 is labeled as “Results and discussion” (line 208), but an additional section 4 as Discussion is included (line 384). Revise this apparent mistake.

Response: We have reorganized and improved the discussion section and deleted the word “and discussion” in the subheading of the section 3.

8. Lines 416-418: why speculate? Any reference to support this statement?

Response: We intended to express the meaning that as the AflR can regulate the distant genes outside the aflatoxin biosynthetic cluster, the BlnR can also regulate the distant genes in addition to the bln gene cluster. In this version of manuscript, we have changed these sentence as “Consistent with these results, our study also showed that BlnR can regulate the genes within the balanol biosynthetic cluster as well as other distant genes involved in many other metabolic pathways (Figure 6 and Table 2)” as shown in the Line461-464.

9. Conclusions are pretty general and even disordered so that authors can include more specific findings from a mechanistic point of view.

Response: Thank you for your kind suggestions. We have improved the conclusions in a mechanistic point of view.

Reviewer 2 Report

The manuscript is devoted to two issues - the study of genes responsible for the synthesis of a biologically active substance, and the study of the influence of the composition of the medium on the yield of this biologically active substance. In my opinion, these two questions in the manuscript are not related. There is no doubt that both parts of the manuscript are interesting and have scientific value, but they do not look logical within the same publication. If the authors consider this study to be complete, then they should have assessed how changing the nutrient medium leads not only to an increase in the yield of the target substance, but also to a change in the expression of genes/enzymes responsible for its synthesis.

The Introduction does not seem logical. The authors describe the enzyme-inhibiting effect of balanol, and further point out that the fungus-producer is used in folk medicine as a means to correct postmenopausal changes. What is the connection between this?

The Discussion covers only the first part of the results concerning the genes responsible for the biosynthesis of balanol. But the part of the study concerning the study of the influence of the environment on the yield of the product remained without discussion.

I also have a few minor remarks:

Please check that all Latin names are correct.

L39 " parasitic fungi mushroom of certain Elaphomyces" - Is it correct?

Figure 8 should be corrected. Brief legends to be deciphered.

Author Response

 Thank you for your valuable suggestions. We will response the comments as the following  one by one:

1. Thank you for your kind suggestion. We think these two issues aimed at high-production of balanol, which contains the regulation-based on part and medium optimization-based on part. We have assessed the transcription level of the gene members within the gene cluster bln after medium optimization and found there is no significant difference (So the data is not shown).

2. In the introduction part, after the enzyme-inhibiting effect of balanol was described, we further pointed out that the fungus-producer is used in folk medicine as a means to correct postmenopausal changes. It’s a general description on the balanol-produced strain, which was attracted as its medicinal history.

3. “The parasitic fungi mushroom of certain Elaphomyces” has been revised into “The parasitic mushroom of certain Elaphomyces”.

4. For Figure 8, we introduced a brief legend and corrected the legend icons.

5. Through the manuscript, we have checked all Latin names carefully.

Reviewer 3 Report

This manuscript reports the genetic engineering of the mushroom fungus Tolypocladium ophioglossoides for the production of balanol, which is a potent protein kinase C inhibitor. This work explored the genes responsible for the regulation of balanol biosynthesis, and the BlnR gene was found to be part of the regulation of the target compound production. This research investigated the cluster-situated Zn2Cys6-family regulator BlnR toward activation of balanol biosynthesis through EMSA experiments. It was found that the BlnR could be involved in the balanol biosynthesis, and it may involve in the development of asexual conidiospore and mycelium growth. With this gene regulation and the study of optimal medium compositions, the highest concentration of balanol was obtained at 2187.39 mg/L. Experiments were well carried out, and this work provides useful information for readers. It is highly recommended for publication after minor revision. In order to improve this manuscript, please consider the comments and suggestions, which are listed below.

  1. Keywords should contain the words “Fermentation”; “Protein kinase C inhibitor”.
  2. Synonyms of Tolypocladium ophioglossoides are Elaphocordyceps ophioglossoides and Cordyceps ophioglossoides; this information should be provided in the introduction.
  3. “Moreover, SM production by microbes is ob-63 served to vary with the composition of culture media and culture conditions [21]. Hence 64 changing culture media is also as a classic approach to activate cryptic gene clusters.”; one fungus can produce many types of secondary metabolites by changes of culture media, please see One strain-many compounds (OSMAC) method for production of polyketides, azaphilones, and an isochromanone using the endophytic fungus Dothideomycete sp., Phytochemistry. 2014;108:87-94; Metabolite diversification by cultivation of the endophytic fungus Dothideomycete sp. in halogen containing media: Cultivation of terrestrial fungus in seawater, Bioorg Med Chem. 2017;25(11):2868-2877. It would be nice to underscore the importance of culture media changes that led to the SM production.
  4. Please re-write this sentence “E. coli cells was cultured in LB broth was used 92 for culture of at 37 °C and 200 rpm.”. Two verbs “was” in one sentence?
  5. “The optimal media was statistically optimized through response..”; please use “medium”, not media. The word “medium” is single, but the word “media” is plural.
  6. “Culture broth were sampled for analysis…”; please use “was”.
  7. “The abaA, fluG and flbD was upregulated..”; please use “were”.
  8. “lysine biosynthesis and aromatic amino acids biosynthesis was up-regulated significantly…”; please use “were”.
  9. After genetic manipulation to produce the desired strain, is the new strain stable? A few sentences discussing on this point would be informative for readers.
  10. Ref 1. Palaniappan Kulanthaivel,Yali F. Hallock, Christie Boros, Sean M. Hamilton, William P. Janzen, Lawrence M. Ballas, Carson R. Loomis, and 453 Jack B. Jiang. Balanol: A novel and potent inbibitor of protein kinase C from the fungus Verticum balanoides, J. A. C. S. 1993,115: 6452-6453. The name of the journal is not correct. Is it J. Am. Chem. Soc.? The format of the references should follow the MDPI style.

Author Response

Thank you for your carefully reading and kind suggestion. Our response to the  comments was as the following one by one:

1. Keywords should contain the words “Fermentation”; “Protein kinase C inhibitor”.

Response: We added the key words Fermentation and Protein kinase C inhibitor in Line 22-23  ?.

2. Synonyms of Tolypocladium ophioglossoides are Elaphocordyceps ophioglossoides and Cordyceps ophioglossoides; this information should be provided in the introduction.

Response: The information about the synonyms of Tolypocladium ophioglossoides is Cordyceps ophioglossoides was included as shown in Line

3.“Moreover, SM production by microbes is ob-63 served to vary with the composition of culture media and culture conditions [21]. Hence 64 changing culture media is also as a classic approach to activate cryptic gene clusters.”; one fungus can produce many types of secondary metabolites by changes of culture media, please see One strain-many compounds (OSMAC) method for production of polyketides, azaphilones, and an isochromanone using the endophytic fungus Dothideomycete, Phytochemistry. 2014;108:87-94; Metabolite diversification by cultivation of the endophytic fungus Dothideomycete sp. in halogen containing media: Cultivation of terrestrial fungus in seawater, Bioorg Med Chem. 2017;25(11):2868-2877. It would be nice to underscore the importance of culture media changes that led to the SM production.

Response: We have underscored the importance of culture media changes on SM production as shown in Line 66-69.

4. Please re-write this sentence “E. coli cells was cultured in LB broth was used 92 for culture of at 37 °C and 200 rpm.”. Two verbs “was” in one sentence?

Response: Thank you for your carefully reading. We have corrected this sentence as “ E. coli cells was cultured in LB broth was at 37 °C and 200 rpm. ”

5.“The optimal media was statistically optimized through response..”; please use “medium”, not media. The word “medium” is single, but the word “media” is plural.

Response: Thank you for your carefully reading. We have corrected this sentence as “ E. coli cells was cultured in LB broth was at 37 °C and 200 rpm. ” in Line 105.

6. “Culture broth were sampled for analysis…”; please use “was”.

Response: Thank you for your carefully reading. As shown in Line 220we have corrected the word “were” as “was”.

7.“The abaA, fluG and flbD was upregulated.”; please use “were”.

Response: Thank you for your carefully reading. We have corrected the word “was” as “were” in the sentence of “The abaA, fluG and flbD was upregulated”.8. 8. “lysine biosynthesis and aromatic amino acids biosynthesis was up-regulated significantly…”; please use “were”.

Response: Thank you for your carefully reading. We have replaced the “was” by “were ” in the above sentence as shown in Line 327.

9. After genetic manipulation to produce the desired strain, is the new strain stable? A few sentences discussing on this point would be informative for readers.

Response: The new strain blnROE is very stable. We have discussed its stability as shown in Line294-296, “And the blnROE strain kept stable for balanol production after 10 generations grown on a PDA plate without selective pressure.”

The insertion of blnR gene into T. ophioglossides was confirmed by PCR,

10. Ref 1. Palaniappan Kulanthaivel,Yali F. Hallock, Christie Boros, Sean M. Hamilton, William P. Janzen, Lawrence M. Ballas, Carson R. Loomis, and 453 Jack B. Jiang. Balanol: A novel and potent inbibitor of protein kinase C from the fungus Verticum balanoides, J. A. C. S. 1993,115: 6452-6453. The name of the journal is not correct. Is it J. Am. Chem. Soc.? The format of the references should follow the MDPI style.

Response: Thank you for your carefully reviewing. We have revised this conference according to the MDPI style.

Round 2

Reviewer 2 Report

The authors have significantly revised the manuscript, and it has become much better. However, I believe that the unreported evidence that environmental optimization does not affect the level of transcription of the genes of the target cluster, but leads to an increase in product yield, should be described in the manuscript. It's really valuable. The authors argue that environmental change can have an impact at different levels, using literature data only. If the authors add their data that the optimization of the environment affects exactly not at the transcriptional level, but perhaps at the level of enzymes or something else, then this will tie together the two parts of the manuscript. Moreover, it will show the future prospects of this study. In this regard, I believe that it is advisable to add these additional data to the manuscript.

Author Response

Response to Reviewer’s comments:

       Here we will express deep gratitude for your deep thinking and kind consideration. In this revised manuscript, we added the comparative analysis of the transcription of gene members within gene cluster bln as shown in Figure 9B. When cultured in the optimized medium, their transcription didn’t change significantly as the effect of overexpression of blnR (as shown in Line410-413 and Line 479-482).